

# Top-down and bottom-up controls on an herbivore on a native and introduced plant in a tropical agricultural landscape

Emma Despland[1] and Paola G. Santacruz[2]

[1] Biology Department, Concordia University, Montreal, QC, Canada
[2] Museo Interactivo de Ciencias, Quito, Ecuador

## ABSTRACT

The recent introduction in a tropical agricultural environment of a weedy open-habitat plant (*Solanum myriacanthum*) and subsequent host range expansion of a common forest-edge butterfly (*Mechanitis menapis*) onto that plant provides an opportunity to examine reconfiguration of tritrophic networks in human-impacted landscapes. The objectives of this study were (1) determine if the caterpillars on the exotic host are more or less limited by plant defenses (bottom-up forces) and if they experience enemy release (decrease of top-down pressure) and (2) define how anthropic open pasture habitat influences the herbivore's tritrophic niche. Field and laboratory monitoring of larval survival and performance on a native (*Solanum acerifolium*) host plant and the exotic (*S. myriacanthum*) host plant were conducted in the Mindo Valley, Ecuador. Plant physical defenses were also measured. Results showed that larval mortality was mostly top-down on *S. acerifolium*, linked to parasitism, but mostly bottom-up on *S. myriacanthum*, possibly linked to observed increased plant defenses. Thus, in the absence of co-evolved relationships, herbivores on the exotic host experienced little top-down regulation, but stronger bottom-up pressures from plant defenses. These findings provide a rare empirical example of enemy-free space as a mechanism underlying host-range expansion. *S. myriacanthum* was less colonized in open pastures than in semi-shaded habitats (forest edges, thickets): fewer eggs were found, suggesting limited dispersal of adult butterflies into the harsh open environments, and the survival rate of first instar larvae was lower than on semi-shaded plants, likely linked to the stronger defenses of sun-grown leaves. These findings show how environmental conditions modulate the rewiring of trophic networks in heavily impacted landscapes, and limit a biocontrol by a native herbivore on an invasive plant in open habitats.

## INTRODUCTION

Changing land use patterns disrupt species' niches, and can lead to new associations (*Agosta, 2006*), especially in the tropics where high biodiversity imposes strong biotic pressures on organisms (*Bonebrake et al., 2010*). These novel trophic relationships that arise by ecological fitting are not tightly co-evolved but emerge as a result of the functional

Corresponding author
Emma Despland,
emma.despland@concordia.ca

traits of species that come in contact with each other (*Agosta, 2006*). The effects of these new community assemblages on insect herbivores are best understood in a tri-trophic perspective, as top-down effects of predators and parasitoids can determine the host plant range of herbivores and play a significant role in defining their niche (*Stireman & Singer, 2018*; *Vidal, Murphy & Scherber, 2018*). Species invasions and changing land use, in particular land-clearing, redefine niches of herbivorous insects via bottom-up and top-down mechanisms. In increasingly human impacted landscapes, species that are able to expand their ranges to include exotic host plants and to colonize open agricultural habitats are less vulnerable to extinction risk (*Despland, 2014*; *Jahner et al., 2011*).

Most tropical herbivorous insects feed on a restricted range of host-plants (*Coley & Barone, 1996*; *Forister et al., 2015*), and thus the insect's spatial distribution and habitat use often depend on the distribution of larval host plants. Indeed, the host plant structures the larval ecology of insect herbivores: it imposes direct bottom-up selection pressures and influences top-down pressure from natural enemies (*Singer et al., 2004*). Exotic plants do not have co-evolved relationships with local herbivores or with the parasitoids and predators on the third trophic level, and novel plant-herbivore associations can show dramatically different outcomes (*Sunny, Diwakar & Sharma, 2015*). In some cases, the lack of co-evolved relationship implies that insect herbivores have no mechanism to counter plant defenses, resulting in lower performance and survival on the exotic host, leading to herbivory release and explaining how an exotic plant can become invasive (*Levine, Adler & Yelenik, 2004*). At the extreme, exotic plants can be evolutionary traps (*Keeler & Chew, 2008*), if they are accepted as oviposition sites by females, but support little or no larval growth. Conversely, exotic plants can provide enemy-free space to herbivores (*Mulatu, Applebaum & Coll, 2004*; *Murphy, 2004*), promoting host range expansion, even if bottom-up pressure on the novel host is stronger (*Lefort et al., 2014*). In this case, native herbivores can provide biocontrol of the exotic plant (*Sunny, Diwakar & Sharma, 2015*). In novel plant-herbivore interactions, the bottom-up pressure from plants can be either greater or less than in co-evolved relationships, but top-down pressure from natural enemies is usually less (*Stireman & Singer, 2018*). In general, performance and survival are lower for larvae developing on exotic hosts relative to native hosts (*Yoon & Read, 2016*).

The interactions between a herbivore and its host plants also depend on plant community composition (*Agrawal, Lau & Hambäck, 2006*). The novel open pasture habitats created by tropical deforestation and agriculture are dominated by weedy light-demanding plants, often including introduced species. The differences between contiguous semi-shaded secondary forest or thicket habitats and open sunny habitats affect both the insect's mobility and the plant's defenses (*Morante-Filho et al., 2016*). Harsh environmental conditions in open pastures can limit dispersal of adult butterflies: *Scriven et al. (2017)* found that less than half of the butterfly species captured in a forest were found to disperse into adjacent open areas, and most of the dispersers used open-habitat plants as larval hosts. While semi-shaded secondary forest, thicket and ecotone habitats can be important biodiversity reservoirs, especially for forest-edge butterflies, open habitats like pastures are used by far fewer species (*Beckmann et al., 2019*;

*Bonebrake et al., 2010*). Moreover, within a plant species, sun-grown individuals are often better defended, with thicker and tougher leaves, more trichomes, and higher concentrations of defensive compounds (*Jansen & Stamp, 1997*; *Kitajima, Wright & Westbrook, 2016*). Overall, the level of herbivore damage to plants in open habitats is often lower than in secondary forest and ecotone habitats due to the above-mentioned mechanisms (*Diaz et al., 2011*; *Jansen & Stamp, 1997*; *Maiorana, 1981*); however, it is sometimes higher due to predator release of open-habitat herbivores (*Coley & Barone, 1996*; *Morante-Filho et al., 2016*).

The recent introduction of a weedy open-habitat plant (*Solanum myriacanthum*) and subsequent host range expansion of a common forest-edge butterfly (*Mechanitis menapis*) onto that plant provides an opportunity to test hypotheses surrounding reconfiguration of tritrophic networks in anthropized environments. Our first objective is to determine how the host range expansion affect bottom-up and top-down pressures on this oligophagous herbivore. *M. menapis* specializes on Solanaceae plants with strong phytochemical and physical defenses; however, on the most common native host plant (*Solanum acerifolium*), mortality seems mostly due to top-down pressure, notably a parasitoid wasp (*Santacruz-Endara, Despland & Giraldo, 2019*). We tested if the caterpillars on the novel exotic host, *S. myriacanthum*, are more or less limited by plant defenses (bottom-up forces) and if they experience enemy release (decrease of top-down pressure). Our second objective is to better define how the creation of open pasture habitats influences this forest-edge herbivore's tritrophic niche. The exotic plant, *S. myriacanthum* spreads invasively in open pastures whereas closely-related native Solanaceae host plants do not (Fig. 1). We examined whether the tri-trophic network operates in the same way in anthropic pastures as in native ecotone habitats, and tested whether herbivory by the butterfly can help control the invasive plant in open pastures.

These two hypotheses were examined through a series of field and common garden experiments. We first surveyed host plant use by *M. menapis* in an agricultural landscape on two native (*S. acerifolium* and *S. candidum*) and one introduced (*S. myriacanthum*) plant. Larvae were reared on the three hosts in an enemy-free common garden, and leaf toughness and trichome density were measured to evaluate bottom-up pressures. Larval survival and performance were next monitored in the field to evaluate top-down forces. Finally, *S. myriacanthum* in ecotone and open habitats were compared in terms of the caterpillar performance they support and their physical defensive traits.

## MATERIALS AND METHODS

### Study species

*Mechanitis menapis* specializes on forest edges (*Young & Moffett, 1979*) and is common in disturbed agricultural landscapes (*Santacruz-Endara, Despland & Giraldo, 2019*). Known host plants are in the Solanum subgenus *Leptostemonum* (*Robinson et al., 2010*), the "spiny Solanums", characterized by sharp epidermal prickles and stellate trichomes (*Levin, Myers & Bohs, 2006*). Caterpillars are gregarious and feed collectively, using silk to avoid plant trichome defenses (*Despland, 2019*; *Despland & Santacruz-Endara, 2016*).

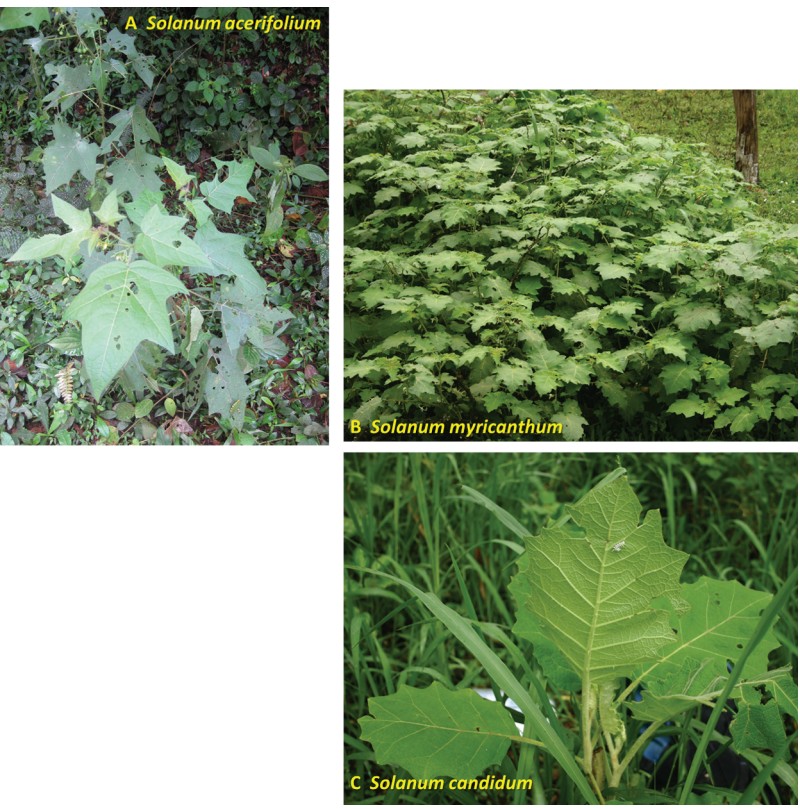

**Figure 1 Host plants used by *M. menapis* in the study region.** (A) *S. acerifolium*, native. (B) *S. myriacanthum*, exotic. (C) *S. candidum*, native. A fourth instar larva is visible on the underside of the *S. candidum* leaf.

The study was conducted in the Mindo valley (00°03′44.1″S 78°45′41.7″W), located in cloud forest at 1,250 m A.S.L. on the Western slope of the Andes in the province of Pichincha, Ecuador. In this region, the main host plant (*Santacruz-Endara, Despland & Giraldo, 2019*) is *S. acerifolium* Dunal sect. *Acanthophora*, subgenus *Leptostemonum* (*Nee, 2019a*). One other known host plant (*Robinson et al., 2010*), *S. candidum* Lindl sect. *Lasiocarpa* within subg. *Leptostemonum* (*Whalen, Costlich & Heiser, 2019*), is also found locally. Both are weedy shrubs of secondary vegetation, roadsides, thickets and agricultural landscapes at moderate altitudes across central and south America (*Nee, 2019a*; *Whalen, Costlich & Heiser, 2019*). *S. myriacanthum* Dunal sect. *Acanthophora*, a weedy shrub of cultivated lands and pastures whose native range spans from Mexico to northern Nicaragua (*Nee, 2019b*), has recently been observed in the Mindo region, and *M. menapis* appears to have expanded its range to include this novel host (*Santacruz-Endara, Despland & Giraldo, 2019*). *S. myriacanthum* uses more open habitats than either *S. acerifolium* or *S. candidum*, including full-sun pastures where it tends to exclude other vegetation (see Fig. 1).

### Field survey
We conducted a field survey of *S. acerifolium*, S. *myriacanthum* and S. *candidum* in ecotone habitats (*n* = 300 plants per species), and of *S. myriacanthum* in open pastures

($n$ = 300 plants) recording the developmental stage of all *M. menapis* individuals seen. The two native host plants, *S. acerifolium* and *S. candidum* were never observed in pasture habitats. Cocoons of the parasitoid *Hyposoter* spp. (Ichneumonidae), a common mortality agent of *M. menapis* in the region (*Santacruz-Endara, Despland & Giraldo, 2019*), were also recorded. Field work was conducted on private land, authorized by land-owners (Ignacio de la Torre; Maria Elena Garzon Jaramillo). Specimens were not collected and no field permit was required.

Numbers of individuals at each stage was compared between the three plants using a GLM with a Poisson error function, after testing for model assumptions. All statistical analyses were done with the R 3.5.3 package (*R Core Team, 2019*).

## Field survival rates

We further monitored the in situ development of *M. menapis* on *S. acerifolium* and *S. myriacanthum* in ecotones (numbers from *S. candidum* were too low to warrant continuing the study) and *S. myriacanthum* in open pastures. Plants ($n$ = 10) with *M. menapis* eggs were flagged in three pastures and in adjoining ecotone habitats, and monitored at 3-day intervals for 1 month (8 observations on each of 90 plants), recording the instar of all observed larvae to reconstruct larval survival. Any apparent causes of mortality were recorded, notably parasitoid cocoons. An instance of parasitism was recorded as the disappearance of a larva and appearance of a parasitoid cocoon between observations. The rate of parasitism was calculated as the proportion of larvae of a given stadium that were parasitized. At each visit, temperature and solar radiation were recorded in each ecotone and pasture location when the sun was out between 10 and 14 h.

Analyses compared between the native *S. acerifolium* and the exotic *S. myriacanthum* in the ecotone, and between ecotone and pasture *S. myriacanthum* plants. Survivorship on the two hosts was compared with Kaplan–Meier survival analysis: to determine the instar at which differences in survival occur, proportions surviving from one developmental stage to the next were compared with chi-square analyses.

## Common garden rearing

Eggs were collected in the field on *S. acerifolium*. Larvae were reared from hatching on potted *S. acerifolium* ($n$ = 80), *S. myriacanthum* ($n$ = 80) and *S. candidum* ($n$ = 20) plants in a common garden, in an indoor insectary with large mesh screen windows to prevent natural enemy entrance. Ten larvae (two groups of five because *M. menapis* are gregarious (*Despland & Santacruz-Endara, 2016*)) were placed per plant. Conditions were similar to those found in ecotone habitats, including semi-shade and natural photoperiod. As in the field monitoring, the instar of all surviving larvae was recorded every 3 days. Mass of all surviving individuals was recorded at pupation with a portable balance (Ohaus Scout SPX123).

As in the field monitoring, survival rates between *S. acerifolium* and *S. myriacanthum* were compared with Kaplan–Meier survival analysis and with chi-square tests for each

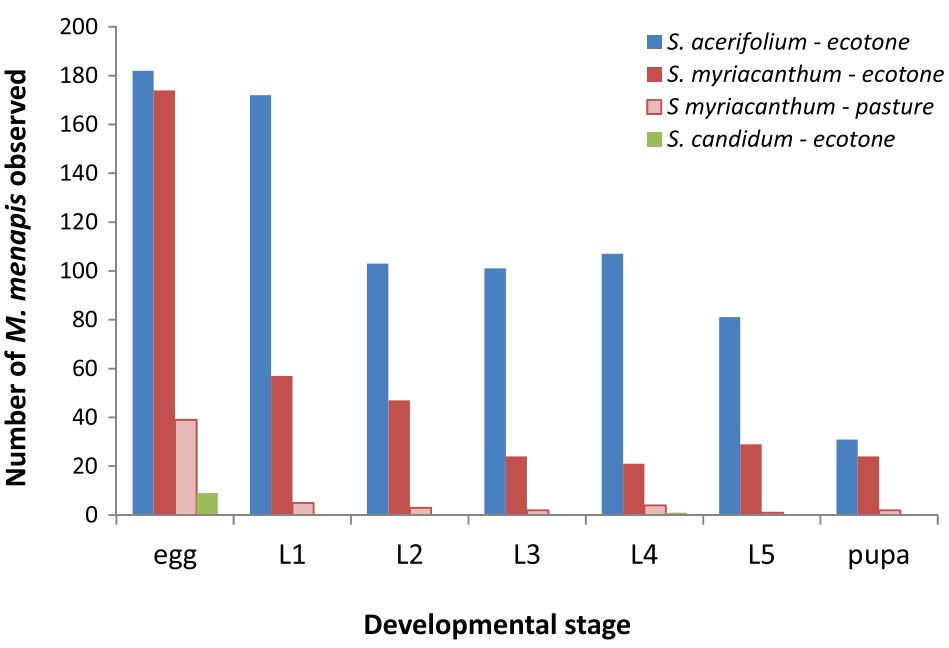

**Figure 2** Number of *M. menapis* eggs, pupae and larvae of each stadium (L1–L5) observed on each of the host plant types in the field survey: *S. acerifolium*, *S. candidum* and *S. myriacanthum* in ecotone habitats and *S. myriacanthum* in pastures.

larval instar. Pupal masses were compared with *t*-tests. Survival on *S. candidum* was too low for inclusion in the analysis.

## Leaf traits

Physical traits were recorded on mid-age leaves (between leaf position three and six from the apex, the leaves on which *M. menapis* are generally found) of field-collected ecotone *S. acerifolium, S. myriacanthum* and *S. candidum* ($n$ = 20 plants per species). The density of stellate, simple and glandular hairs on 4 mm$^2$ leaf discs was counted under a stereomicroscope (Nikon Fabre Photo EX microscope, 20× magnification). For each leaf, three discs were punched in the proximal, medial and distal thirds of the leaf, avoiding secondary and tertiary veins, and pooled to create an average value per leaf.

Specific leaf area (SLA) was evaluated on 45 mm diameter leaf discs, avoiding the midvein, recording fresh and dry mass with the portable balance to calculate water content. Leaf toughness was evaluated as the force to fracture the leaf lamina using a penetrometer (*Cobo-Quinche et al., 2019*). Trichome density, water content and SLA were also measured on open habitat *S. myriacanthum*.

Leaf traits were compared between the three plant species and between ecotone and open-habitat *S. myriacanthum* using GLMs with the appropriate error function.

## RESULTS

### Field survey

Of the 300 plants surveyed in ecotone habitats, 182 eggs were observed on *S. acerifolium*, 174 on *S. myriacanthum*, and only 9 on *S. candidum* (see Fig. 2). Data from the native

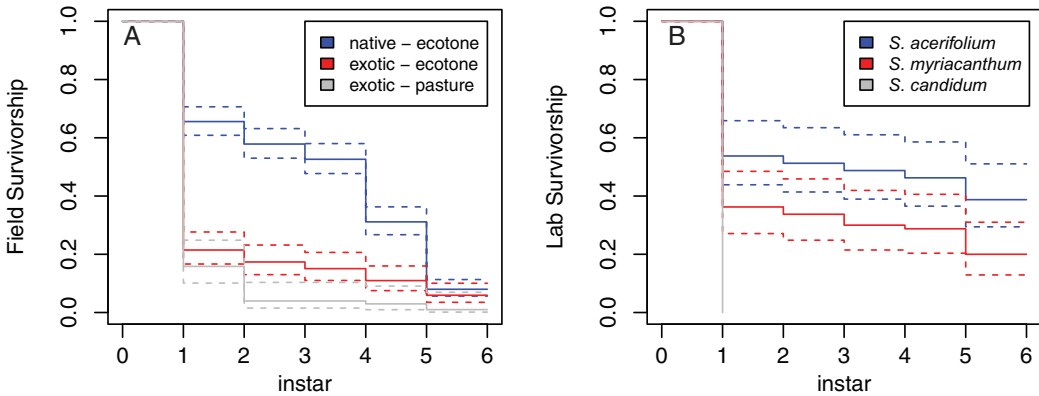

**Figure 3 Survivorship curves for *M. menapis* larvae on the various host plants tested.** (A) Native *S. acerifolium* and exotic *S. myriacanthum* in ecotone habitats and *S. myriacanthum* in pasture habitats. (B) *S. acerifolium*, *S. myriacanthum* and *S. candidum* in the common garden experiment.

*S. acerifolium* and the exotic *S. myriacanthum* only were included in the analysis, due to the very low numbers on the native *S. candidum*. Numbers observed differed significantly between the two host plants ($z_{598} = 0.906$, $p < 0.0001$), and interaction terms suggested differential survival between host plants at several developmental stages (first instar: $z_{598} = 2.97$, $p = 0.002$; second instar $z_{598} = 3.04$, $p = 0.002$; fourth instar $z_{598} = 2.15$, $p = 0.03$; fifth instar $z_{598} = 1.94$, $p = 0.043$). Notably *Hyposoter* parasitoid pupae ($n = 72$) were only observed on *S. acerifolium*.

On *S. myriacanthum*, more eggs were observed on ecotone plants than on pasture plants ($n = 174$ vs. $n = 39$; $\chi^2 = 131$, $p < 0.0001$). Numbers of larvae were not compared due to the extreme difference in initial numbers of eggs. No parasitoid pupae were seen in either habitat type.

## Field survival rates

Kaplan–Meier survivorship curves for native *S. acerifolium* and exotic *S. myriacanthum* in ecotones, as well as *S. myriacanthum* in pastures are shown in Fig. 3. Survival analysis showed significant differences: *S. acerifolium* vs. *S. myriacanthum* in ecotones, $z = 7.59$, $p < 0.0001$; *S. myriacanthum* in ecotone vs. pasture, $z = 2.07$, $p = 0.038$.

Chi-square analysis showed that, in ecotone habitats, survival of first instar larvae was higher on native *S. acerifolium* than on exotic *S. myriacanthum*, but survival late in development was higher on *S. myriacanthum* (Table 1). Rate of *Hyposoter* parasitism was high on *S. acerifolium* (15% for fourth instar larvae and 37% for fifth instar larvae) but non-existent on *S. myriacanthum*, potentially explaining the difference in survival rate. Indeed, when parasitized insects were removed from the analysis, the differences in mortality rates in instars 4 and 5 lost significance (survival on *S. acerifolium* in instar 4 = 0.86; $z = 0.87$; $p = 0.32$; in instar 5 = 0.80; $z = 1.11$; $p = 0.15$).

On *S. myriacanthum*, survival of first instar larvae was higher in the ecotone than in the pasture habitat, but survival rates at subsequent instars did not differ significantly, and overall survival did not differ significantly between habitat types (see Table 1).

**Table 1 Field survival rates at each larval instar on native *S. acerifolium* vs. exotic *S. myriacanthum* in ecotones, as well as exotic *S. myriacanthum* in ecotone vs. pasture habitats.**

| Field survey | Instar 1–2 | Instar 2–3 | Instar 3–4 | Instar 4–5 | I 5–pupa | Overall |
|---|---|---|---|---|---|---|
| Native | **0.66** | 0.88 | 0.92 | **0.61** | **0.32** | 0.11 |
| Exotic | **0.21** | 0.81 | 0.87 | **0.83** | **0.72** | 0.09 |
| Rate $\chi^2$ | **104** | 1.29 | 0.61 | **4.01** | **8.48** | 0.49 |
| *p*-Value | **(<0.0001)** | (0.25) | (0.42) | **(0.04)** | **(0.004)** | (0.67) |
| Ecotone | **0.21** | 0.81 | 0.87 | 0.83 | 0.72 | 0.09 |
| Pasture | **0.16** | 0.81 | 0.93 | 0.81 | 0.65 | 0.06 |
| Rate $\chi^2$ | **4.22** | 2.44 | 2.62 | 1.31 | 1.50 | 3.28 |
| *p*-Value | **(0.04)** | (0.12) | (0.11) | (0.26) | (0.22) | (0.07) |

Note:
Chi-square with one degree of freedom and *p*-values are given—significant values are shown in bold.

**Table 2 Laboratory survival rates at each larval instar on native *S. acerifolium* and exotic *S. myriacanthum*.**

| Lab rearing | Instar 1–2 | Instar 2–3 | Instar 3–4 | Instar 4–5 | I 5–pupa | Overall |
|---|---|---|---|---|---|---|
| *S. acerifolium* | 0.61 | 0.96 | 0.96 | 0.95 | 0.86 | 0.45 |
| *S. myriacanthum* | 0.36 | 0.93 | 0.89 | 0.96 | 0.78 | 0.22 |
| Rate $\chi^2$ | **9.03** | <0.001 | 0.42 | <0.001 | 0.02 | **7.12** |
| *p*-Value | **(0.003)** | (0.98) | (0.51) | (0.99) | (0.89) | **(0.008)** |

Note:
Chi-square with one degree of freedom and *p*-values are given—significant values are shown in bold.

Both temperature and luminosity were considerably higher in the pasture than in the ecotone environment (31.6 ± 1.6 s.d. °C vs. 23.4 ± 1.68 s.d. °C, 117,000 lux ± 4,600 vs. 69,000 ± 11,000 lux respectively).

## Common garden performance

In the common garden, survival was highest on the native *S. acerifolium*, lower on the exotic *S. myriacanthum* (Kaplan–Meier survival analysis $z = 2.6$; $p = 0.009$), and zero on the native *S. candidum* ($z = 4.91$; $p < 0.0001$). Chi-square analysis showed that survival of first instar larvae was higher on *S. acerifolium* than on *S. myriacanthum*, but that survival at later developmental stages did not differ between the two host plants (see Table 2). None of the larvae reared on *S. candidum* survived beyond the first instar.

Pupal mass did not differ significantly between insects reared on *S. acerifolium* and *S. myriacanthum* (167 mg ± 8. S.E. ($n = 37$) vs. 160 mg ± 13 S.E. ($n = 20$); *t*-test $t_{56} = 0.54$; $p = 0.5$).

## Leaf traits

The trichome profile differed between the three plants tested: *S. candidum* had much higher numbers of stellate trichomes on both leaf surfaces (GLM with quasipoisson link function: $t_{54} = 12.2$; $p < 0.0001$); *S. myriacanthum* had more simple trichomes (GLM with

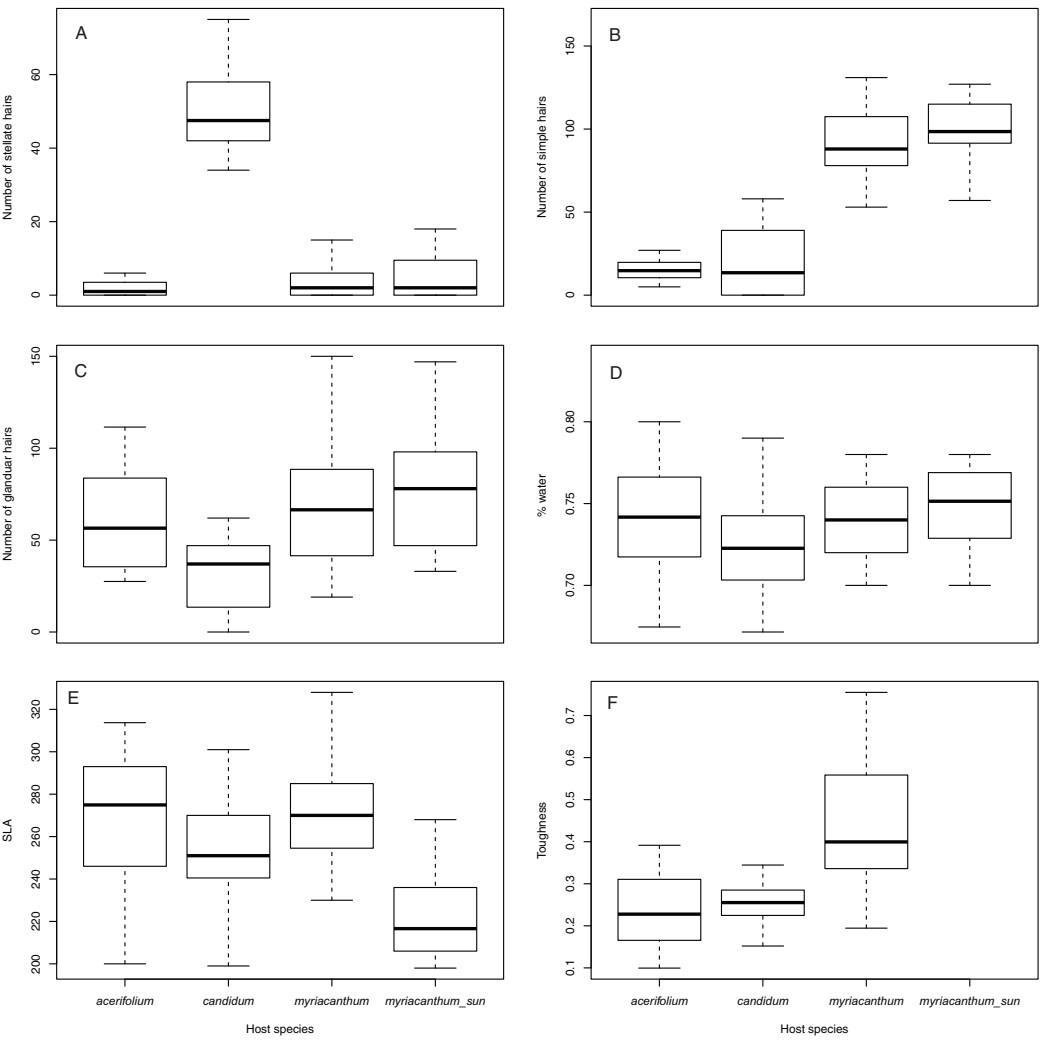

**Figure 4 Number of trichomes and leaf traits on all three host plants in ecotones, as well as on *S. myriacanthum* in open habitats.** Traits include: (A) stellate, (B) simple, (C) glandular hairs, (D) water content, (E) SLA and (F) toughness. Toughness data for open habitat *S. myriacanthum* is unfortunately not available.

quasipoisson link function: $t_{54} = 16.5$; $p < 0.0001$) and glandular trichomes (GLM with quasipoisson link function: $t_{54} = 2.10$; $p = 0.004$) on the abaxial surface. No difference was recorded in SLA ($F_{2, 57} = 2.02$, $p = 0.14$) or in water content ($F_{2, 57} = 1.52$, $p = 0.228$) between the three species. However, leaves of *S. myriacanthum* were significantly tougher ($F_{2, 57} = 20.52$, $p < 0.0001$) than those of *S. acerifolium* or *S. candidum* (see Fig. 4).

*Solanum myriacanthum* growing under full sun showed a greater number of stellate (GLM with quasipoisson link function: $t_{36} = 3.04$; $p = 0.02$), simple (GLM with quasipoisson link function: $t_{36} = 3.02$; $p = 0.03$), and glandular trichomes (GLM with quasipoisson link function: $t_{36} = 1.90$; $p = 0.04$)—see Fig 4. SLA was significantly higher in ecotone than in open-area leaves ($t_{38} = 46$; $p < 0.0001$), but water content did not differ ($t_{38} = 0.90$; $p = 0.35$).

## DISCUSSION

In response to the first objective, results suggest that bottom-up pressure increased, but top-down control decreased on the exotic, relative to the native, host plant. Indeed, both common garden experimental and field observational results showed different patterns of mortality between *S. acerifolium* and *S. myriacanthum*. In the field, mortality on *S. acerifolium* occurred mostly in the late larval instars, and seemed mostly due to parasitism by *Hyposter*. By contrast, mortality on *S. myriacanthum* occurred mostly early in development (in both the common garden experiment and in the field), and parasitism was never observed. Removing the effect of parasitoids (by common garden rearing in an insectary or by post-hoc manipulation of field data) led to higher survival on the native than on the exotic plant. Thus, population control of *M. menapis* on *S. acerifolium* appears mostly top-down, linked to parasitism, whereas limiting factors on *S. myriacanthum* appear more bottom-up, possibly linked to plant defenses.

The high mortality of first instar *M. menapis* on *S. myriacanthum* is possibly linked to a higher density of simple and glandular trichomes, and to higher toughness compared to the closely related, native *S. acerifolium*. Indeed, glandular trichomes are effective defenses against early-instar *M. menapis*, limiting the ability of small caterpillars to initiate a feeding edge and establish themselves on the leaf (*Despland, 2019*). Phytochemistry likely also plays an important role (*Beccaloni, 1995*), but measuring plant chemical defenses was beyond the scope of this study. Overall, our results suggest that the *M. menapis* host range has expanded to include *S. myriacanthum;* this exotic plant appears to provide enemy-free space, and thus to become a viable host despite strong defenses—for a similar example see *Murphy (2004)*. Herbivores on chemically defended plants, like the Solanums, can experience a trade-off between host plants with low defenses that support good growth but provide low potential for sequestration of plant compounds as defense against natural enemies, and highly defended plants that support low growth but provide enemy-free space (*Mira & Bernays, 2002*; *Murphy & Loewy, 2015*; *Zalucki et al., 2012*).

The second native host studied, *S. candidum*, did not support survival in the common garden experiment and was very seldom used in the field. *S. candidum* is listed as a *M. menapis* host plant (*Robinson et al., 2010*), but clearly is very marginal in our study region. *Mechanitis* is a species complex in which larval host plant use is an important taxonomic trait; however, most information on host plant use comes from anecdotal records, and does not adequately represent frequency or geographical range of host use, obscuring a clear interpretation of host use patterns (*Giraldo & Uribe, 2012*). Our results suggest possible genetic differences, in the plant or in the butterfly, between our study region and those where this relationship was observed.

In response to the second objective, results show that open pasture conditions limit the herbivore's expansion onto the exotic host. Fewer *M. menapis* eggs were found on *S. myriacanthum* plants in pastures and the survival rate of first instar larvae was lower than on ecotone plants. Low oviposition in full sun can arise from butterfly preference for partially shaded habitats. Adult *M. menapis* were never seen in pastures in the course of

the study. Harsh environmental conditions thus appear to play an important role in limiting *M. menapis* dispersal into pastures (*Scriven et al., 2017*). Low first instar survival could be linked to the observed higher trichome density and lower SLA (generally a good proxy for greater toughness) of full sun plants. Indeed, within a species, sun leaves are often tougher and bear more trichomes than shade leaves (*Kitajima, Wright & Westbrook, 2016*). Leaves of several *Solanum* species have been shown to be tougher, and to exhibit lower SLA, more trichomes and more allelochemicals when grown in full sun than in partial shade, and the specialist caterpillar *Manduca sexta* shows lower performance on sun-grown *Solanum* plants (*Jansen & Stamp, 1997*). Similarly, herbivores perform better on shade than on sun leaves of *Solanum viarum* sect. *Lasiocarpa*, a sister species to *S. myriacanthum;* and, by consequence, plants in shade habitats show more herbivore damage (*Diaz et al., 2011*).

The novel trophic relationship between *M. menapis* and *S. myriacanthum* is thus modulated by habitat, demonstrating how trophic relationships can reconfigure depauperized communities in heavily disturbed landscapes: in this system, the native host plants are restricted to semi-shade secondary vegetation thickets and ecotone habitats. The arrival of an exotic weed that tolerates the harsh conditions in full sunlight can lead to its rapid proliferation in pastures. Herbivore pressure on the invasive plant is low in pastures, which become herbivore-free space, perhaps facilitating the plant's spread. Similarly, the invasion of the closely-related *S. viarum* in Florida has shown how weedy plants can exhibit different growth patterns and biomass allocation in pastures than in partially-shaded habitats and can spread dramatically in the absence of top-down herbivore control, becoming noxious weeds excluding other vegetation (*Diaz et al., 2014*).

## CONCLUSIONS

Our study shows how both plant species invasions and novel habitat creation via land-clearing for agriculture can rewire trophic relationships between the native forest-edge species that dominate tropical agricultural landscapes.

First, our findings support the paradigm that, in the absence of a co-evolved relationship, bottom-up pressure from plant defenses is stronger on exotic hosts, but that herbivores experience less top-down control on these exotic plants, which can ameliorate their value as hosts (*Mulatu, Applebaum & Coll, 2004*; *Murphy, 2004*). The tri-trophic niche can therefore facilitate native herbivore host range expansion onto exotic plants (*Stireman & Singer, 2018*), and exotic plants can become a valuable resource for insect conservation (*Despland, 2014*; *Jahner et al., 2011*).

Second, however, our findings also show that pasture habitat conditions limit colonization of an exotic plant by a native herbivore, and that this herbivore is therefore of little use as a biocontrol agent on a weedy exotic plant that is invading these anthropic habitats. Full-sun pastures in the tropics are harsh microhabitats relative to forest-edges, and generally exhibit low diversity, and are vulnerable to becoming overwhelmed by a few weedy, often exotic, species to the exclusion of other organisms. Thus, although secondary vegetation and thickets can be important biodiversity reservoirs for tropical forest-edge

species, pastures constitute a harsh environment that is much less used (*Horner-Devine et al., 2003*).

## ACKNOWLEDGEMENTS

Thanks to Tania Iza for help with rearing and field work, and to Rafael Cardenas for help with toughness measurements.

### Funding

This research was funded by a Fonds de recherche du Québec–Nature et Technologies (FRQNT) échanges hors Québec de professeurs award (grant # FRQ-NT 211156). The funders had no role in study design, data collection and analysis, decision to publish, or preparation of the manuscript.

### Grant Disclosures

The following grant information was disclosed by the authors:
Fonds de recherche du Québec–Nature et Technologies (FRQNT) échanges hors Québec de Professeurs Award: FRQ-NT 211156.

### Competing Interests

The authors declare that they have no competing interests.

### Author Contributions

- Emma Despland conceived and designed the experiments, performed the experiments, analyzed the data, prepared figures and/or tables, authored or reviewed drafts of the paper, and approved the final draft.
- Paola G. Santacruz performed the experiments, authored or reviewed drafts of the paper, and approved the final draft.

### Field Study Permissions

The following information was supplied relating to field study approvals (i.e., approving body and any reference numbers):
Field work was conducted on private land, authorized by land-owners (Ignacio de la Torre; Maria Elena Garzon Jaramillo). Specimens were not collected and no field permit was required.

### Data Availability

The raw data is available in the Supplemental Files.

### Supplemental Information

Supplemental information for this article can be found online at http://dx.doi.org/10.7717/peerj.8782#supplemental-information.

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
