# Peer review of "Top-down and bottom-up controls on an herbivore on a native and introduced plant in a tropical agricultural landscape"

_PeerJ, doi:10.7717/peerj.8782_

## Round 0.1 · original submission · Major Revisions

The reviewers and I agree that your study has much merit, although the manuscript does not do it justice. There are various presentation issues as well as the need to pay close attention to the evidence in relation to the interpretations and conclusions. Both reviewers have done an excellent job of detailing these issues and suggesting ways to address them. Please address each of their comments and use their suggestions as guidelines for revisions.

Reviewer 1 ·

Basic reporting

This is an interesting paper examining the performance of a native herbivore on native and exotic host plant species. The main finding is that on the exotic host, herbivore mortality is higher early in life. In contrast, on native host plants, herbivore mortality is higher on later instars, which was attributed to parasitism. Although the question is important for the study of insect-plant interactions, the manuscript needs substantial improvement.
Overall the paper doesn’t read well, therefore the authors should work on the flow. To facilitate the understanding, it would be important to keep reminding readers that S. myriacanthum is exotic and that S. acerifolium is native. This is particularly important for tables and figures, as well as, the discussion section.

Experimental design

It is stated that the role of habitat in defining the herbivore’s tritrophic niche was tested, however, I do not see how this was tested

Validity of the findings

The main conclusion stated in the paper is that the exotic plant is ameliorating the negative effects of pastures by interacting with a caterpillar. However, all the evidence provided showed that herbivores avoided open areas and performed poorly on plants from open habitats. Therefore, several conclusions are equivocal.

Something to take into consideration is that top-down control is largely influenced by plant traits. Predators and parasitoids would have a hard time finding their prey/host if the plant is sending mix messages. Therefore, the lack of parasitism on exotic hosts may be attributed to host quality, plant quality or plant signals. I am also curious about whether this Ithomiine butterfly sequesters plant defenses. Several Ithomiine species can sequester alkaloids from plants, so is it possible that on exotic hosts they were not parasitized because they were more toxic?

Additional comments

Introduction
Overall the introduction is very confusing, several topics are presented, but they didn’t seem match the actual results. It is stated that the role of habitat in defining the herbivore’s tritrophic niche was tested, however, I do not see how this was tested. If habitat is really being tested than I am concerned with replication because only one ecotone habitat and one pasture were surveyed. How big was the area sampled? How far apart were the plants?
L80: How or why was this species introduced? Wouldn’t this be a colonization event?
L51-52: why would forest loss benefit herbivores and not natural enemies? Is it because herbivores arrive first in the new environment? Please explain
L86-88: What are the hypotheses/predictions?
L80-93: Authors should clearly state their hypotheses and how they were they tested. I was unsure about how the authors tested the effects of habitats, especially because only one plant species occurs in both habitats. One alternative to clarify the hypotheses is to have them followed by the information in lines 94-102.
L105: why unlike most Ithomiine butterflies? These butterflies thrive in old field environments

Methods
Is it possible to correlate plant defenses and herbivore performance? It’s unclear which and how many plants were used for leaf trait analyses. If leaf traits were assessed on the same plants where caterpillars were found, then correlations could be made, and the discussion would be less speculative.
L121-123: Is there any data that would support this argument?
L126: Does this means that 600 plants were surveyed for S. myriacanthum? How many plants actually had eggs/caterpillars on them?
L130-131: Why repeated measures? From the description it sounded like 300 plants per sp were screened, but it didn’t mention that plants were revisited. If a repeated measure analyses was used were the plants marked and revisited? How do you know you are sampling the same individual? Were plants and caterpillars marked?
L133: italicized in situ
L136: Instar?
L139-140: was the habitat temperature and solar radiation recorded per habitat or plant?
L141-143: A GLM with logit distribution would be more adequate
L152: Why was survival analysis not used?
L158-160: Was the position of leaf discs standardized? How leaf was treated in the analyzes? It is stated that 10 leaves per plant was used, but from how many plants? Were the leaves pooled to create one value per plant? This is one reason why it is important to provide degrees of freedom for all statistics.

Results
S. myriacanthum was less colonized (fewer eggs) in open habitats compared to ecotone habitats. S. acerifolium: no comparison between open and ecotone habitats. Caterpillar mortality is attributed to parasitism, but the statistical comparison is not presented. When parasitized caterpillars are removed from the analysis, is mortality different?

L170-173: Please present your F-statistics and degrees of freedom. P-values are not informative.
L173-174: In both habitats?
L175-176: how many plants per habitat were inspected? If more plants are found in ecotone habitats, I assume more eggs will be found there as well, so how was this controlled in the analysis?
L181-183: Where is this result? No figure or table presented this information. Also, when only unparasitized larvae are taken into consideration, was the survival rate different?
L191-195: This result suggests that plant defenses might not be important
L195: What is the SE? Please provide this information for all results
L201: SLA means specific leaf area, so you can delete “leaf”

Discussion
Start your discussion by stating the purpose of the study and the questions and what was the key finding of the study.
L214: What is an important taxonomic trait?
L 220: it is unclear whether parasitism doesn’t occur because parasitoids avoid hosts feeding on S. myriacanthum or just because that plant species harbors lower host density.
L221: What do you mean by population controls?
L226-235: Murphy & Feeny’s paper (PNAS and Ecological monographs) should be included in the discussion of host shift and enemy space.
L236-238: But was this result controlled by the number adults? If adults are not found in pastures, then fewer eggs are expected in pastures as well. This result doesn’t seem surprising, unless there is a difference between the native host. The authors need to discuss alternative hypotheses for this result. Is the habitat effect similar for S. acerifolium?
L247-249: Confusing, please revise.
L250-252: It is unclear how this novel interaction is facilitated by the habitat. This seems an equivocal conclusion. The results showed that a reduced number of eggs and lower herbivore performance in open habitats. Therefore, I don’t understand how this can be beneficial for disturbed landscapes.
L262-264: What kind of edge species? Plants or herbivores? This study did not test the ability of species to colonize pastures.

Table 1: Provide SE and Chi-square value
Table 2: What is overall? Survival from hatching to pupa? Only 9% and 6% of the caterpillars survived?
Fig: 2- It is stated that 300 plants were observed. Were all the plants in the same habitat?
Fig. 3 - Post hoc test comparisons needed. Why was toughness data for open habitat not available?

Reviewer 2 ·

Basic reporting

Many sentences were not in professional English and many were structured poorly. For these reasons, this paper was difficult to follow. I attempted to rewrite some sentences and provide example sentences to make the paper read more clearly and with less ambiguity. I have also provided comments asking for additional information.

The authors have sufficient references and background relating to this study. However, the authors could improve the paper by adding a couple sentences defining a host plant and what is meant by the term “novel” (see line 61). I also suggest rewriting the sentence on lines 71-72 to be clearer, see my comment.

The article structure was good. However, I have provided edits and feedback for the tables and figures.

Overall, the paper is meaningful but needs to be written more clearly and in professional English.

Experimental design

I believe this paper has merit for the journal, but I recommend a statistician to review the analyses and results. The research questions should be restated to be clearer (I attempted to rewrite the objectives in the abstract and introduction to give the authors a starting point). The materials and methods section is well organized and detailed. However, the authors should state what statistical software program was used and what alpha (significance level) was set to (i.e. 0.05)—see comments within this section.

Validity of the findings

Findings are rationale and beneficial to literature pertaining to trophic relationships as well as plant-insect interactions. The authors do a good job relating their finding to the study objectives in the discussion and conclusion section. Again, I suggest having a statistician review the analyses and results prior to accepting the paper into the journal.

Additional comments

Suggestions to improve Figures and Tables:

Figure 1: Scientific names imbedded in the photos need to be italicized and ‘photo credit’ needs to be stated in the figure as well. See figure and table referencing in the journal guidelines.

Figure 2: Good figure! Replace the word “stadium” with “stage” and define L1, L2, L3, etc. in the caption (similar to my comments for Tables 1-3). Also, italic the species names within the figure. Overall, the table is clear and can be easily interpreted.

Figure 3: Replace “#” with “Number of” for the appropriate y-axes. Missing a period after the “S” in the species scientific name in the last sentence.

Tables 1: Add a note below the table or in the table description that clearly explains what L1 to L2, L2 to L3, … represents. I also suggest stating L1 to L2 as “Instar 1-2”. Tables should be stand alone and some readers who are not entomologist may not understand the larvae development. Also, add the “rates” to the left of the p-values and put the p-values in parentheses. Please see my comments associated with the table.

Tables 2 and 3: I strongly suggest making the same revisions discussed about Table 1 to Tables 2 and 3.

Annotated reviews are not available for download in order to protect the identity of reviewers who chose to remain anonymous.

---

## Round 0.2 · Minor Revisions

Your revision has addressed the major issues based on reviewer comments. In the interest of efficiency, I have added edits and comments to the Word file of the revised manuscript (see annotated ms). Please address these points, which are all minor. One other point concerns figure 2. Revise the legend as follows "...each stage (L1-L5) observed..." As to figure 2 itself, the different shades in greyscale are difficult to discern. I suggest changing one of the dark grey shades to black.

---

## Round 0.3 · accepted · Accept

Your edits all look good. Nice work!

I looked at the figures in greyscale with the idea that some readers might not see color as well as others. It's not necessary, but I like to check it from that perspective.

I have also passed along your request for a low-income country fee waiver to the PeerJ staff.